# YUSEG:Yolo and Unet is all you need for cell instance segmentation

**Bizhe Bai**
University of Queensland
Australia
bizhe.bai@outlook.com

**Jie Tian**
Southern Medical University
Guangzhou, Guangdong, China
salen@i.smu.edu.cn

**Sicong Luo**
Xi'an Jiaotong University,
Xi'an, Shaanxi , China
lsc19980723@foxmail.com

**Tao Wang**
Fuzhou University
Fuzhou, Fujian, China
1072739254@qq.com

**Sisuo LYU**
Harbin Institute of Technology, Shenzhen
Shenzhen, Guangzhou, China
sisuolv@outlook.com

## Abstract

Cell instance segmentation, which identifies each specific cell area within a microscope image, is helpful for cell analysis. Because of the high computational cost brought on by the large number of objects in the scene, mainstream instance segmentation techniques require much time and computational resources. In this paper, we proposed a two-stage method in which the first stage detects the bounding boxes of cells, and the second stage is segmentation in the detected bounding boxes. This method reduces inference time by more than $30\%$ on images that image size is larger than 1024 pixels by 1024 pixels compared to the mainstream instance segmentation method while maintaining reasonable accuracy without using any external data.

## 1 Introduction

Cell instance segmentation, which recognizes individual cell bodies in a microscope image, is helpful for cell analysis. For instance, quantitative cell biology requires measurements of a wide range of cellular parameters, including form, location, RNA expression, and protein expression. Before researchers can assign these attributes to particular cells in the image, they must segment an image into cell instances. However, due to the high computational cost caused by a large number of objects in the scene, using mainstream instance segmentation methods, for example, Mask R-CNN[1] and Cascade R-CNN[2] directly on large-size microscopes, is slow[3].In addition, those methods could not do instance segmentation on whole slide images (WSI) directly because of the limitation of GPU memory [4]. More specifically, the COCO dataset, whose majority of images are about 640 pixels by 480 pixels, is used for training and inference by mainstream instance segmentation methods [5]. However, this image size is considerably smaller than WSI, which has a size of about 3000 pixels by 3000 pixels. Thus, we proposed a two-stage method that combines YOLO and Unet for microscopes instance segmentation that reduces inference time by employing a more lightweight object detection network and reduces GPU memory consumption by training and inference on window-sliding patches.YUSEG maintains the same accuracy as the mainstream instance segmentation method while reducing inference time by more than 30 percent for images larger than 1024 pixels by 1024 pixels. I will introduce the outline of the YUSEG method at the beginning of section 2 and explain the detail of the YUSEG method in the rest of section 2. Then I will state details of the YUSEG training and inference process in section 3 and the result and discussion in section 4. Lastly, I will conclude section 5.

36th Conference on Neural Information Processing Systems (NeurIPS 2022).

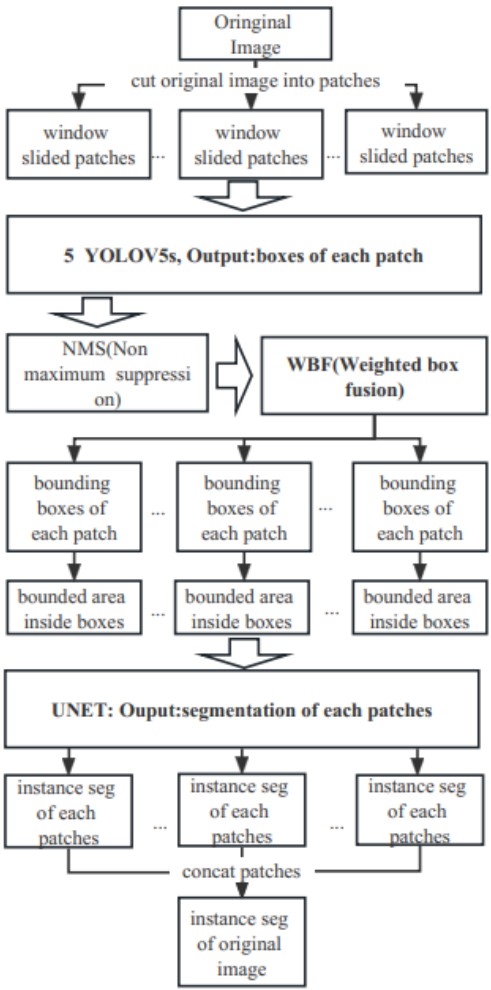

Figure 1: The architecture of the YUSEG instance segmentation method. The inference process has two stages. In the first stage, images larger than 1024 pixels by 1024 pixels will be cut into smaller patches. Then, each patch will be fed into a model that is an ensemble of five YOLOV5[6] to predict the bounding boxes of each cell. Then, non-maximum-suppression and weighted box fusion are applied to those bounding boxes to remove duplicated detected boxes. Next, in the second stage, the area of the original input image, which is bounded by the bounding boxes, will be fed into a semantic segmentation model, which has been trained on training images, to separate pixels into cells and the background. In the end, the segmentation result will be put where the bounding boxes are, and the patches will be concatenated to make the final instance segmentation result of the original input image.

## 2 Architecture of YUSEG

YUSEG is a two-stage method combining an object detection model, an ensemble of five yolov5[6] models and a segmentation model based on Unet[7]. The illustration is depicted in figure1.In the training round, two models need to be trained separately: (1) the YOLOV5[6] object detection model and (2) a semantic segmentation neural network based on Unet[7] with the efficientnet[8] encoder. The object detection model seeks to detect the bounding boxes for each cell in images, whereas the semantic segmentation model seeks to distinguish cells from the background. Next, I will describe the preprocessing technique we used in Section2.1, then describe the object detection model in section 2.2, and then describe the semantic segmentation model in section 2.3.

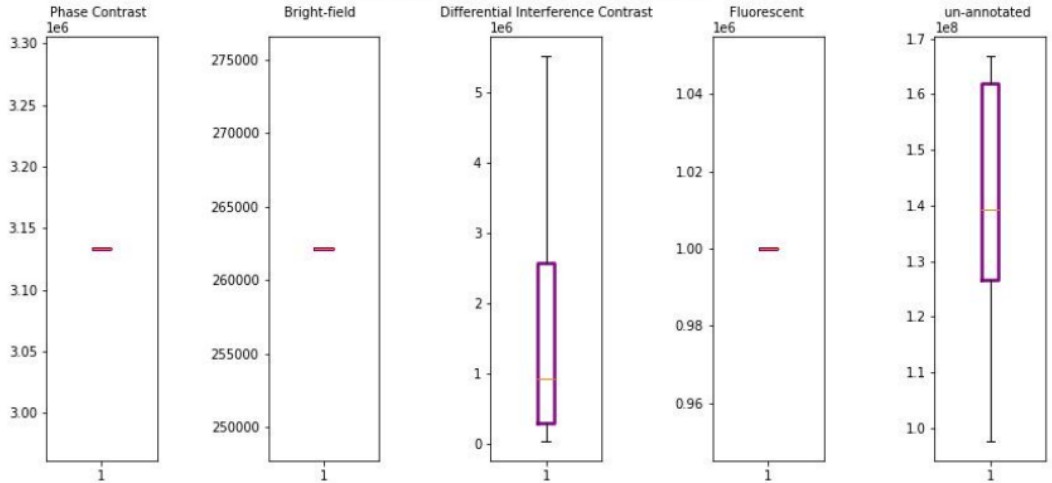

Figure 2: Box plot of the number of pixels in the training set. From left to right are plots of the number of pixels for phase contrast modality, differential interference contrast modality, brightfield modality, and fluorescent modality. The rightmost boxplot represents the number of unlabeled pixels.

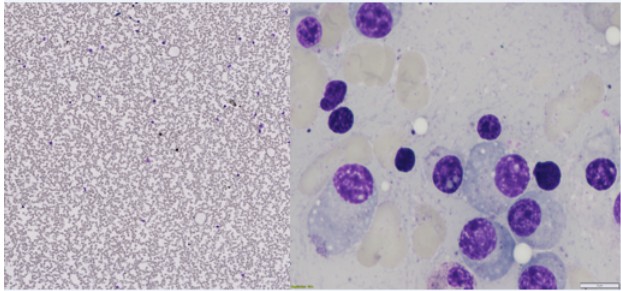

Figure 3: The illustration of a simple resize strategy. The original size of the image on the left is 4096 pixels by 4096 pixels, while the image on the right is 480 pixels by 640 pixels. If the left image were resized to the same size as the right image, it would be difficult to distinguish individual cells.

## 2.1 Preprocessing

The dataset contains four types of images: phase-contrast, differential interference contrast, brightfield, and fluorescent. A straightforward approach would be to feed the training data directly to the CascadeRCNN[2] model. Nonetheless, as shown in Figure 2, some images with large sizes exist. For instance, the number of pixels in a phase contrast image is approximately $3 * 10^6$, which is too large to feed directly into an algorithm due to the memory limitations of modern graphic cards. Thus, some works tend to resize all images to a single size before feeding them to a network[9]. However, resizing the size of a larger image would result in the loss of information. As depicted in Figure 3, resizing the larger image, which is on the left, to the same size as the smaller image, which is on the right, will raise the difficulty of distinguishing cells in the images. Using sliding window patched-based training and inference enables the model to accommodate the dataset's diverse sizes of input images. In the YUSEG architecture, the input images will be divided into multiple 1024 pixels by 1024 pixels patches and fed to the object detection model. During inference, the images will also be divided into numerous patches of the same size. This method would preserve more information in the larger image than a straightforward resizing strategy.

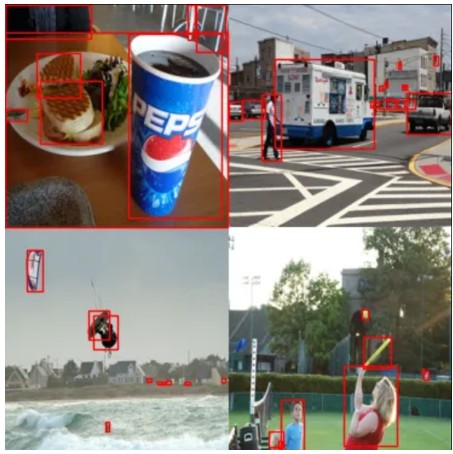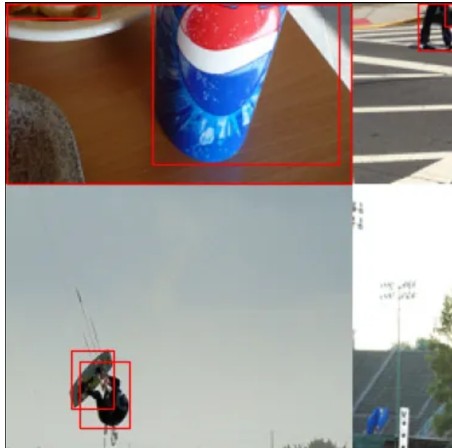

Figure 4: Left image is a combination of four images. The right image is after Mosaic data augmentation of four images in the object detection part. Four images will be randomly zoomed, cropped, arranged, and spliced into a new image. Each picture will have its corresponding bounding boxes.[12]

Table 1: Comparison between an ensemble of five YOLOV5 models and single YOLOV5 model on tunning set

| | Ensemble of five models | Single model |
|---|---|---|
| **Mean F1** | 0.8122[1] | 0.7966 |

## 2.2 Object Detection stage

The object detection model is based on YOLOV5. It is a combination of five YOLOV5 models. Each YOLOV5 model will be trained independently on $80\%$ of training data sampled at random during the training period. Ensembling five models significantly raise the mean F1 Score on the tuning set as shown in Table1. In the inference period, each YOLOV5 model predicts its bounding boxes. Since there are five YOLOV5 models jointly predicting bounding boxes, duplicate or overlapping boxes will exist. Therefore, non-maximal suppression is applied to the bounding boxes to eliminate the duplicated boxes introduced by the five YOLOV5 models. Then, weighed box fusion is applied to bounding boxes to remove duplicated cells between overlapping patches. Next, I will introduce several techniques we applied.

**Mosaic Data augmentation**
Normally, the metrics of small targets are much lower than that of medium and large targets. The microscope dataset also contains a large number of small targets. Thus, we embedded Mosaic data augmentation [10]. Mosaic is inspired by the CutMix[11] data augmentation proposed at the end of 2019, compared to CutMix only uses two images for splicing, while Mosaic data enhancement uses 4 images, randomly zoomed, randomly cropped, and randomly arranged for splicing. Mosaic data augmentation uses four images to stitch images, and each image has its corresponding bounding boxes. After splicing the four pictures, a new picture is obtained, and the boxes corresponding to the picture are also obtained. Then the new picture is passed into the neural network, equivalent to passing in four pictures at a time for learning. This greatly enriches the background of detected objects. The augmented data will be computed during the batch normalization computation, which will also benefit the neural network. An example of mosaic data augmentation is shown in figure 4

---

[1]This metric is reported by the competition organizers and shown on the official leaderboard of the website

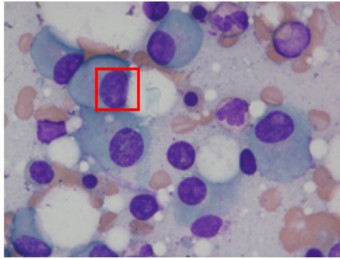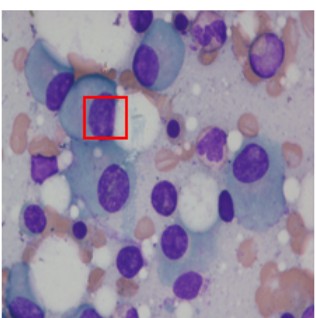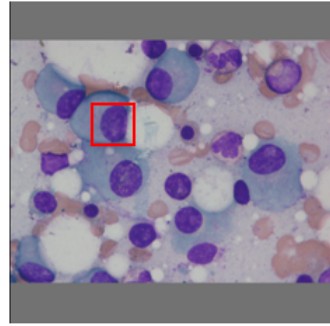

Figure 5: Visualization with adaptive image scaling and padding. The left image is the original image, the middle image is a simple resizing strategy using bilinear interpolation, and the right image results from adaptive scaling and filling. We can observe that the ratio between the height and width of the right cell is the same as the original. However, the cells inside the red box in the middle image have been stretched.

**Adaptive image scaling and filling**

Besides large-sized images, such as the left image, which has a size of 4096 pixels by 4096 pixels, there are also some small-sized images, such as the right image, which has a size of 480 pixels by 640 pixels. However, the input size of YUSEG is manually set to 1024 pixels by 1024 pixels in consideration of computational cost. Therefore, smaller images must be resized to 1024 pixels by 1024 pixels. However, simple resizing strategies, such as bilinear interpolation, cause the cell's aspect ratio to change, as shown in Figure 5. Therefore, we added adaptive image scaling and filling in YUSEG and adaptively added the least black border while maintaining the original aspect ratio to preserve the original image as much as possible. The steps are as follows:

(1) The original image size is 800*600, and the target zoom size is 416*416. After dividing the target zoom size by the size of the original image, the two zoom factors are 0.52 and 0.69.

(2) Choose a smaller scale factor. The length and width of the original image are multiplied by the minimum scaling factor of 0.52, then the width becomes 416, and the height becomes 312.

(3) 416-312=104 Calculate the height that needs to be filled, and then use np.mod in numpy to take the remainder to get 8 pixels, then divide it by 2 to get the value that needs to be filled at both ends of the image height.

**Loss function**

The Intersection of Union (IoU), also known as the Jaccard Index, is the most prevalent evaluation metric in object detection benchmarks. IoU has a plateau and cannot be optimized in the case of non-overlapping shapes(or bounding boxes)[13]; specifically, if bounding box A does not intersect bounding box B, i.e., $|A \cap B| = 0$, then $IoU(A, B) = 0$. In this case, IoU will not indicate whether two shapes are close together or far apart. Therefore, Generalized Intersection over Union (GIoU) is chosen as the loss function by YUSEG. As depicted in the diagram, GIoU is the gradient problem that reduces IOU loss when the bounding boxes do not overlap, as shown in Figure 6. The formula of GIOU is :

$$GIoU = 1 - IoU(A, B) + \frac{|C - A \cup B|}{|C|}$$

**Weighted boxes fusion**

Since the YUSEG method is based on patch training and inference, there will be overlap between each patch, so there will be many overlapping bounding boxes of cells in the overlapping area. Therefore, we need to remove those overlapping cells because cells are not allowed to be stacked together horizontally. A straightforward approach to removing duplicate boxes is Non-Maximum Suppression (NMS). However, in some cases, the prediction bounding boxes are all wrong. In this case, NMS will leave only one inaccurate box, while WBF[15] will fuse it using all predicted boxes. The differences are shown in figure 7. So YUSEG embeds WBF into its architecture.

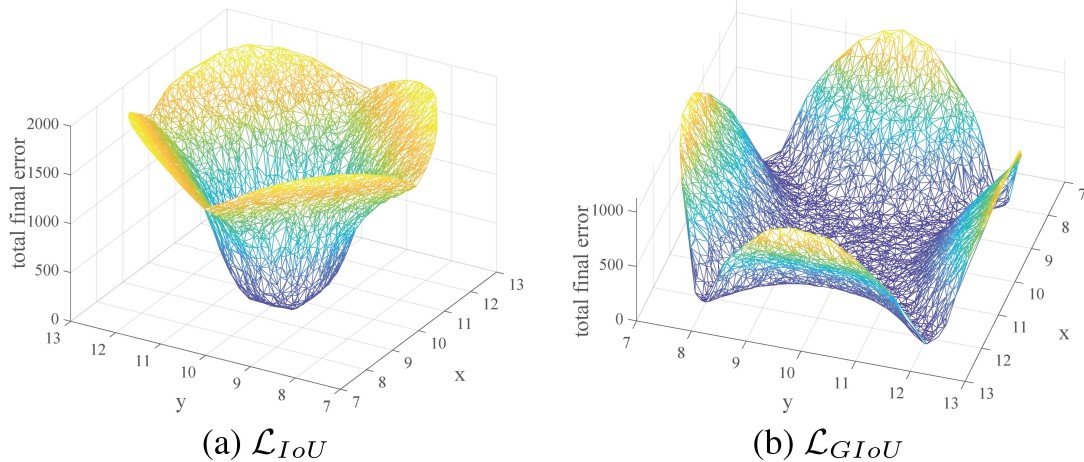

(a) $\mathcal{L}_{IoU}$          (b) $\mathcal{L}_{GIoU}$

Figure 6: The visualization of regression error of IoU and GIoU. One can see that IoU loss has large errors for non-overlapping cases. GIoU loss alleviates the gradient issue of IOU loss when the detection frames do not overlap[14].

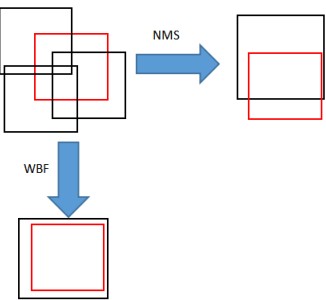

Figure 7: Visualization illustration of NMS and WBF outcomes for an ensemble of inaccurate bounding boxes. Blue boxes are several models' predictions, and red is the ground truth. .[15].

## 2.3 Semantic segmentation of cell

After obtaining the bounding box of each cell, the next step is to distinguish the area of each cell from other areas (other areas, including the background and other cells). YUSEG chose a semantic segmentation method based on the Unet [7] architecture with an efficient [8] backbone. The model structure diagram of Unet is shown in 8. During the training phase of the semantic segmentation model, the area bounded by the outer rectangle of each cell serves as the semantic segmentation model's input. The label of the model is a mask image of the corresponding region, with the target cell labelled 1, the surrounding cells labelled 2, and the background labelled 0 as shown in 9. During the training, the model does a three-class segmentation, centre cell, other cells, and background. The signal from other cells does benefit the model. There is a $3.4\% miou$ increase compared to binary segmentation on the validation set, as shown in Table 2. During the inference procedure, the model will predict masks for three classes and assigns other cell pixels to the background.

Table 2: MIoU on the validation set between three-class segmentation model, which is used in YUSEG, and binary segmentation model.

| | Three class segmentation model | Binary class segmentation model |
|---|---|---|
| **miou** | 85.3% | 88.7% |

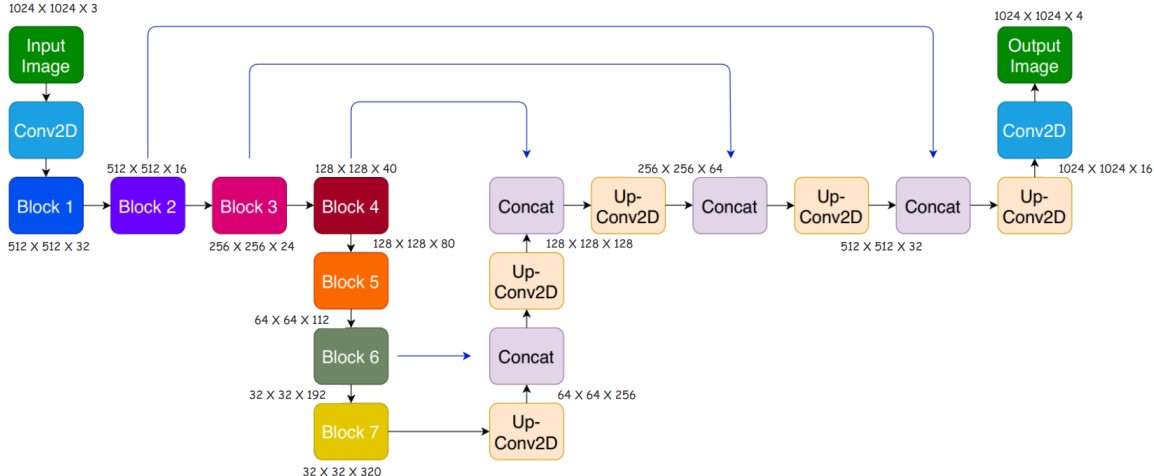

Figure 8: Architecture of EfcientUNet with EfcientNet-B0 framework for semantic segmentation. [16]

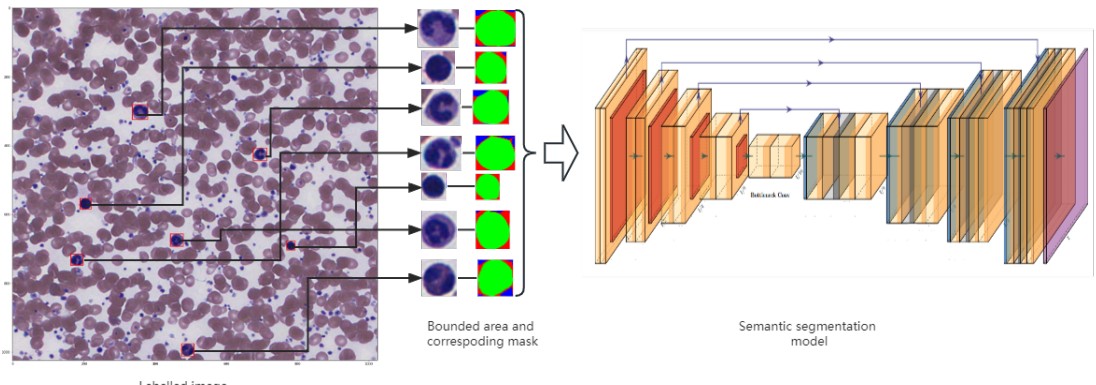

Figure 9: Visualization of the segmentation model training procedure. The area bounded by ground truth bounding boxes and the corresponding ground truth mask will be treated as input and labelled for the semantic segmentation model during training. In this image, the central cell is labelled green, the surrounding cells are labelled blue, and the background is labelled red.

**Techniques to improve speed**

The original YOLOV5[6] architecture writes detection results as text files directly to the disk, including bounding boxes and corresponding images. In the second stage of YUSEG, however, the semantic segmentation model must read these images and bounding boxes from the disk, which requires significant I/O time. Therefore, YUSEG directly embeds the segmentation model into the YOLO architecture. This engineering advancement reduces inference time by more than 30 percent as shown in Table 6. Converted Pytorch checkpoint files to Torchscript checkpoint files are used to reduce inference time further.

# 3 Experiments

## 3.1 Dataset

We do not use any public dataset and pre-trained models.

## 3.2 Implementation details

### 3.2.1 Environment settings

The development environments and requirements are presented in Table 3.

Table 3: Development environments and requirements.

| System | Ubuntu 18.04.5 LTS |
|---|---|
| CPU | CPU Intel Xeon W 2150B @ 3.00GHz |
| RAM | RAM 32x4GB2.67MT/s |
| GPU (number and type) | GPU One Nvidia RTX5000 16G |
| CUDA version | 11.0 |
| Programming language | Python 3.9 |
| Deep learning framework | Pytorch (Torch 1.10, torchvision 0.2.2) |
| Specific dependencies | opencv-python , Pillow , timm ,scipy ,ensemble_boxes ,segmentation_models_pytorch |
| Code | `https://github.com/baibizhe/semi_cell.git` |

### 3.2.2 Training protocols

**Data augmentation**

In the object detection stage, patch-based training and inference strategy is used. We cut each image into 1024 pixels by 1024 pixels patches (slide window with a patch size $1000 \times 1000$).In the segmentation training and inference, the input images are resized to $224 \times 224$.

Table 4: Detection training protocols.

| Network initialization | [17]section 3.3 |
|---|---|
| Batch size | 8 |
| Patch size | $1000 \times 1000$ |
| Total epochs | 50 |
| Optimizer | Adam with momentum 0.9 |
| Initial learning rate (lr) | 0.001 |
| Lr decay schedule | halved by 200 epochs |
| Training time | 16 hours |
| Loss function | GIOU |

Table 5: Segmentation training protocols.

| Batch size | 64 |
|---|---|
| Input size | $224 \times 224$ |
| Total epochs | 500 |
| Optimizer | Adam with momentum 0.99 |
| Initial learning rate (lr) | 0.001 |
| Lr decay schedule | CosineAnnealingWarmRestarts[18] |
| Training time | 12 hours |
| Loss function | Cross-entropy |

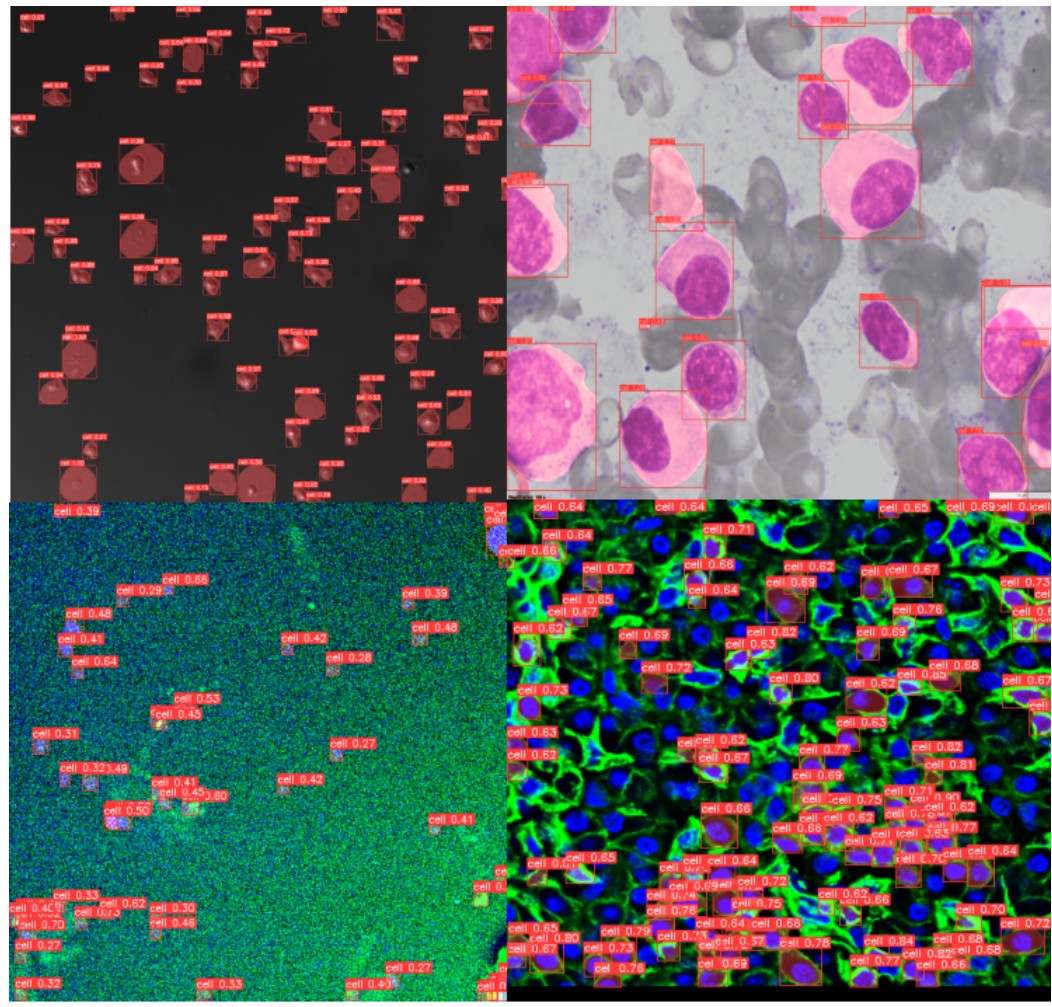

Figure 10: The top two images are good results. The bottom two images are the bad result.

# 4 Results and discussion

## 4.1 Quantitative results on tuning set

The F1 score on the tuning set is 0.8122. We did not use unlabelled data.

## 4.2 Qualitative results on validation set

The visualization of our result on tunning set is shown in Figure 10

## 4.3 Segmentation efficiency results on tunning set

This section shows a comparison between YUSEG and cellpose[19], a mainstream cell instance segmentation method.

Table 6: Comparison of inference time[3] between ours (YUSEG) and Cellpose[19]. The optimal inference time is in bold in each row.

| Tunning set image name | Image Size(pixel) | YUSEG official(s)[4] | YUSEG local(s)[5] | Cell pose(s) [6] | Difference(s) [7] |
|---|---|---|---|---|---|
| cell_00001.tiff | 480, 640 | 10 | 9.2 | **9** | 0.8 |
| cell_00003.tiff | 480, 640 | 10.1 | 9.2 | **7.9** | 1.3 |
| cell_00043.png | 1024, 1024 | 16.3 | 14.2 | **13** | 1.2 |
| cell_00051.png | 1024, 1024 | 17.4 | 15.2 | **14.1** | 1.1 |
| cell_00028.tiff | 944, 1266 | 11.3 | **10.7** | 17.9 | 7.2 |
| cell_00030.tiff | 944, 1266 | 11.3 | **10.6** | 17.2 | 6.6 |
| cell_00071.tif | 2048, 2048 | 14.1 | **13.3** | 44.4 | 31.1 |
| cell_00073.tif | 2048,2048 | 13.9 | **12.8** | 44.6 | 31.8 |
| cell_00004.png | 3000, 3000 | 14.3 | **13.3** | 121.1 | 107.8 |
| cell_00011.png | 3000, 3000 | 14.2 | **13.3** | 126.6 | 113.3 |

Table 7: Final result on the testing set. The metrics are reported on four different modalities separately

|  | All | Brightfield | Differential interference contrast | Fluorescent | Phase-contrast |
|---|---|---|---|---|---|
| Median F1 | 0.7334 | 0.8143 | 0.7777 | 0.3423 | 0.8066 |
| Mean F1 | 0.6321 | 0.7669 | 0.7374 | 0.3814 | 0.6954 |

## 4.4 Results on final testing set

The result of the final testing set is reported on Table 4.4. And the discussion related to the final result is stated in the next section. 4.4

## 4.5 Limitation and future work

First, the generalization of the YUSEG model to fluorescence modal data is inadequate. The mean F1 score for fluorescence test data is 0.3814, whereas the mean F1 score for all data is 0.8066, as shown in Table 4.4. Domain adaptation techniques can improve the generalizability of models. Second, the YUSEG model has no efficiency advantage for small image sizes, such as 480 pixels by 480 pixels images, for which the inference time is comparable to or slower than Cellpose. A potential solution is using a single detection model instead of five to reduce inference time. Nevertheless, a single detection model can negatively affect detection accuracy. Due to unlabeled data, self-supervised training will improve the accuracy of a detection model[20]. Finally, there is an explicit limitation of YUSEG when cells are crowded, as depicted in the two images at the bottom of Figure10. This may be due to the loss of global information for patch-based training [21]. This will be mitigated by adding a global embedding layer to the detection method.

---

[2]The "inference time" represents running the python script for a single image, which includes the initialization time, and importing modules time. This inference time is expected to be longer than the author of Cell pose reported [19].

[3]The "inference time" represents running the python script for a single image, which includes the initialization time, and importing modules time. This inference time is expected to be longer than the author of Cell pose reported [19].

[4]YUSEG official represents the inference time reported by competition organizers which are tested on their machine. The detail of the running time of all images is in appendix5

[5]YUSEG local represents the running time tested on author's local computer with a single RTX3090 graphic card and Intel(R) Core(TM) i5-10600KF .

[6]Running time of cell pose is tested on author's local computer as the same as "YUSEG local".

[7]For a fair comparison, this difference is between "YUSEG local" and "Cell pose" which are tested on a same computer.

# 5 Conclusion

In this paper, we proposed a two-stage instance segmentation method that does not use any external data. The model combines an object detection method and a segmentation model that reduce inference time by more than 30 % compared to the mainstream instance segmentation method while achieving reasonable accuracy. The IOU on the tuning dataset of our algorithm is 0.812.

## Acknowledgement

The authors of this paper declare that the segmentation method they implemented for participation in the NeurIPS 2022 Cell Segmentation challenge has not used any private datasets other than those provided by the organizers and the official external datasets and pre-trained models. The proposed solution is fully automatic without any manual intervention.

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

# Appendices

| Img Name | Real Running Time(s) |
|---|---|
| cell_00001.tiff | 10.08222556 |
| cell_00002.png | 12.61770606 |
| cell_00003.tiff | 10.17986417 |
| cell_00004.png | 14.31007218 |
| cell_00005.png | 14.43264747 |
| cell_00006.png | 12.58284855 |
| cell_00007.tiff | 9.884434938 |
| cell_00008.tiff | 9.838415861 |
| cell_00009.png | 16.87398124 |
| cell_00010.png | 13.23057628 |
| cell_00011.png | 14.22071385 |
| cell_00012.png | 14.30344296 |
| cell_00013.tiff | 10.1827383 |
| cell_00014.tiff | 10.06591034 |
| cell_00015.png | 13.92795444 |
| cell_00016.tiff | 10.13160753 |
| cell_00017.png | 14.5036037 |
| cell_00018.tiff | 10.28819752 |
| cell_00019.tiff | 9.945495129 |
| cell_00020.tiff | 10.1600759 |
| cell_00021.tiff | 10.14462137 |
| cell_00022.tiff | 9.980204105 |
| cell_00023.tiff | 10.27207613 |
| cell_00024.tiff | 10.05274844 |
| cell_00025.png | 12.86001468 |
| cell_00026.png | 12.36447835 |
| cell_00027.tiff | 10.79306173 |
| cell_00028.tiff | 11.3337481 |
| cell_00029.tiff | 11.25484157 |
| cell_00030.tiff | 11.10871959 |
| cell_00031.tiff | 12.21810937 |
| cell_00032.tiff | 12.00715542 |
| cell_00033.tiff | 12.10899806 |
| cell_00034.tiff | 11.7545228 |
| cell_00035.tiff | 11.23081565 |
| cell_00036.tiff | 11.69526792 |
| cell_00037.tiff | 11.51898766 |
| cell_00038.tiff | 13.24681568 |
| cell_00039.tiff | 12.33923984 |
| cell_00040.png | 16.67677188 |
| cell_00041.png | 14.60842705 |
| cell_00042.png | 14.34258342 |
| cell_00043.png | 16.33421373 |
| cell_00044.png | 21.971524 |
| cell_00045.png | 21.92329669 |
| cell_00046.png | 19.71089768 |
| cell_00047.png | 21.20953321 |
| cell_00048.png | 23.91611671 |
| cell_00049.png | 18.06005478 |
| cell_00050.png | 21.01982355 |
| cell_00051.png | 17.42934585 |
| cell_00052.png | 18.23688793 |
| cell_00053.png | 20.74845886 |
| cell_00054.png | 25.40423679 |

| | |
|---|---|
| cell_00055.png | 22.07082558 |
| cell_00056.png | 25.85584092 |
| cell_00057.png | 20.24230123 |
| cell_00058.png | 20.87508655 |
| cell_00059.png | 23.97440982 |
| cell_00060.png | 23.02444124 |
| cell_00061.png | 24.84559822 |
| cell_00062.png | 24.91869354 |
| cell_00063.png | 17.26922679 |
| cell_00064.png | 21.32472754 |
| cell_00065.png | 24.24272537 |
| cell_00066.png | 16.88613272 |
| cell_00067.png | 18.67011094 |
| cell_00068.png | 18.55694175 |
| cell_00069.png | 18.83313823 |
| cell_00070.png | 14.51741123 |
| cell_00071.tif | 14.05365109 |
| cell_00072.tif | 13.56979704 |
| cell_00073.tif | 13.933213 |
| cell_00074.tif | 76.77126098 |
| cell_00075.tif | 10.42145991 |
| cell_00076.tif | 10.01617932 |
| cell_00077.tif | 12.0289185 |
| cell_00078.tif | 12.13260889 |
| cell_00079.tif | 10.29756212 |
| cell_00080.tif | 10.17286038 |
| cell_00081.tif | 10.54105926 |
| cell_00082.tif | 10.29177833 |
| cell_00083.tif | 10.62970424 |
| cell_00084.tif | 10.27098584 |
| cell_00085.tif | 10.22331142 |
| cell_00086.tif | 10.14668727 |
| cell_00087.tif | 10.2383194 |
| cell_00088.tif | 9.951846838 |
| cell_00089.tif | 10.33135796 |
| cell_00090.tif | 10.26525617 |
| cell_00091.tif | 10.09399915 |
| cell_00092.tif | 9.972382784 |
| cell_00093.tif | 9.968453884 |
| cell_00094.tif | 10.02007294 |
| cell_00095.tif | 9.781136751 |
| cell_00096.tif | 10.30164504 |
| cell_00097.tif | 10.17512536 |
| cell_00098.tif | 10.03810811 |
| cell_00099.tif | 19.03290176 |
| cell_00100.tif | 17.54822397 |
| cell_00101.tif | 5964.819666 |

