# OpenReview forum: "YUSEG: Yolo and Unet is all you need for cell instance segmentation"
_NeurIPS.cc/2022/Challenge/CellSeg — Submitted to NeurIPS CellSeg 2022_

### Official Review · Reviewer_QEga · 2022-12-26
**Incomplete and unclear sentences.**

**Rating:** 4
**Confidence:** 3

**Review:**

The paper ensembles five YOLOv5 to detect cells and uses WBF to remove nested boxes. The obtained patches are fed into the UNet model for cell segmentation. The paper is difficult to read and has a lot of spelling errors and word spacing errors. Too many commas, Incomplete and unclear sentences. Please read your submission carefully before submission.

Method:
*) For the YOLOv5, why did you use five models? The more, the better? Is there a reason for designating five?

*) Even if the neighbor cell is very close to the center cell, does it still work well? Based on what criteria did the center cell and neighbor cell be divided?

*) Figure 7 is unclear. Make sure to accurately express what each black box represents (It looks like the black box was written in a blue box)


Experiments:
*) Please show the difference between the existing method and the proposed method in numbers.


Results and discussion:
*) Please summarize 4.3 (Segmentation efficiency results on validation set) and write it sincerely. If there is a new method you suggested, please fill it out sincerely compared to the previous method.

---

> ### Author Response · Authors · 2023-02-21
> **Response to Reviewer QEga**
>
> Thanks for your comment. I will address your concerns point to point.
>
>
> 1. "For the YOLOv5, why did you use five models? The more, the better? Is there a reason for designating five?"
> Yes. Five YOLOv5 is better than one model.I've added comparison between five models and a single model in section "Object Detection stage".
>
>
> 2. "Even if the neighbor cell is very close to the center cell, does it still work well? Based on what criteria did the center cell and neighbor cell be divided?"
> Yes.YUSEG will work.Sorry for confusion I brought. The segmentation model in YUSEG is a three-classes semantic segmentation model.If you are asking how the semantic segmentation model distinguish the center cell and other cells. The first answer is the label we provide to model is three-classes. And model will learn how to distinguish the center cell and other cells.
>
>
> 3. " Figure 7 is unclear. Make sure to accurately express what each black box represents (It looks like the black box was written in a blue box)"
> Thanks for your comment .Figure 7 is deleted since it could not represent what I mean. And I've added a formula and works to describe GIoU better in the chapter "loss function".
>
>
> 4. "Experiments: *) Please show the difference between the existing method and the proposed method in numbers."
> Thanks for your comment . The difference between the existing method, Cellpose, and YUSEG have been added to section 4.
>
>
> 5.  "Results and discussion: *) Please summarize 4.3 (Segmentation efficiency results on validation set) and write it sincerely. If there is a new method you suggested, please fill it out sincerely compared to the previous method."
> Thanks for your comment. I've compared an existing method, Cellpose, and YUSEG in section 4.3.

---

### Official Review · Reviewer_GMMC · 2022-12-26
**Yolo and Unet for Cell Instance Segmentation**

**Rating:** 6
**Confidence:** 4

**Review:**

## Summary
In order to improve the efficiency of multi-instance segmentation, the author proposes a two-stage detection-segmentation model. In the target detection stage, multiple YOLOv5 models were integrated with WBF to fuse the target boxes. The instance segmentation stage uses the UNet which uses EfficientNet as the encoder.

## Methodology
1. Mosaic data augmentation, which spells 4 images into 1. It can enrich the background information of the sample.
2. Adaptively resize the YOLO initial anchor box based on the training set.
3. Object detection uses GIOU LOSS, which is better optimized than IOU LOSS when the predicted box and the true box do not intersect.
4. Only the labeled data provided by the competition is used. Public datasets, pre-trained models, and unlabeled data are not used.

## Comments
1. It would be better to try other models related to UNET in segmentation stage, or make some structural improvements.
2. Part 4.3 gives the inference time of each image segmentation of the verification set, but there is no comparison with other methods, so it is hard to know how fast the inference is.

## Quality
The method proposed in this paper reduces the inference time by 50% compared with the mainstream method while the accuracy is greater than or equal to that of the mainstream instance segmentation method.

## Clarity
1. Steps 2 and 5 of adaptive sizing of the initial anchor box are not explained very clearly. With regard to "Take the remainder to get 8 pixels", who takes the remainder from whom? How does "Using K-means clustering to get n anchor frames" determine n anchor frames, and what is the relationship between k and n?
2. UNET does not give the model structure diagram or the parameter table of each layer.
3. The inferred time is 50% less than the mainstream instance segmentation method ". However, it does not specify the specific method, nor does it give a comparison with other specific methods.

## Conclusion
1. The main contribution of this paper is to improve the inference speed of multi-instance segmentation.
2. The motivation was to solve the problem of slow inference caused by too many examples in a single picture, but the experimental results were not good enough on the picture with multiple small targets.

---

> ### Author Response · Authors · 2023-02-21
> **Response to  Reviewer GMMC**
>
> Thanks for your suggestion. I will address your concerns about this paper point to point.
>
>
> 1.  "UNET does not give the model structure diagram or the parameter table of each layer." \
> Thanks for your suggestion, the model structure diagram has been added in section "Semantic segmentation of cell" or  Figure 8.
>
> 2. "The inferred time is 50% less than the mainstream instance segmentation method ". However, it does not specify the specific method, nor does it give a comparison with other specific methods."  \
> Thanks for your suggestion. The comparison of inferred time between YUSEG and Cellpose[1] is discussed in Table 5.
>
>
> 3. "Steps 2 and 5 of adaptive sizing of the initial anchor box are not explained very clearly. With regard to "Take the remainder to get 8 pixels", who takes the remainder from whom? How does "Using K-means clustering to get n anchor frames" determine n anchor frames, and what is the relationship between k and n?". \
> Thanks for your suggestion. However, after re-training and testing, we found "Adaptive anchor box calculation" technique does not improve the model's performance. Thus, we deleted the explanation of the "Adaptive anchor box calculation".
>
>
> 4. "It would be better to try other models related to UNET in the segmentation stage, or make some structural improvements."
> Thanks for your suggestion. Yes, I agree with you it would be better if we make some structural improvements to the network. But a simple unit with reasonable metrics would benefit our use of our open-source code. More specifically,  our Unet model could be created by only one line of python code. \
>
>
> 5. "Part 4.3 gives the inference time of each image segmentation of the verification set, but there is no comparison with other methods, so it is hard to know how fast the inference is."  \
> Thanks for your suggestion. I've added a comparison between an existing method, Cellpose, and YUSEG in section 4.3.
>
> [1]. Stringer, Carsen, et al. "Cellpose: a generalist algorithm for cellular segmentation." Nature methods 18.1 (2021): 100-106.

---

### Official Review · Reviewer_p9mz · 2022-12-26
**YOLOv5 and UNet for Instance Segmentation of Cells**

**Rating:** 4
**Confidence:** 4

**Review:**

### Summary and Contributions
The authors propose a two-stage instance segmentation pipeline (detection and segmentation) to extract regions of single cell from five different modalities of microscopy images (phase-contrast, bright field, DIC, fluorescent, and un-annotated). The authors claim the two-stage pipeline reduces inference time and GPU memory compared to standard segmentation methods (i.e. Cascade R-CNN by Cai et al. and Masked-Attention Mask Transformer by Cheng et al.).

### Strengths
- Good description of methods
- Clear diagram in Figure 1 (though minor typos) that illustrate the main idea of the proposed methods

### Weaknesses
- Figures 3, 4 and 5 seem to be from other papers - please reference them properly.
- Figure 6 is unclear in what it is trying to convey.
- (Very minor) Proper LaTeX for GIoU equation (use \cup for union, for example)
- (Very minor) Please write out what NMS and WBF stands for before using the acronym, e.g. Weighted Box Function (WBF). This makes the paper easier to read for others.
- Section 3 on Experiments and Section 4 on Results are very lacking in detail (especially Results).
- Overall, it is hard to verify whether the proposed model improves on existing models and the paper does not do a sufficient job in demonstrating the capabilities with respect to existing methods.
- There is a clear limitation of the method when cells are crowded - a discussion of this in Limitations and proposal on how to overcome such problems would strengthen the paper.

### Clarity
- As another reviewer noted, many of the claims of the paper need to be substantiated. As a first draft the paper does well in explaining the proposed methods, but will need significant improvement in explaining its results.

---

> ### Author Response · Authors · 2023-02-22
> **Response to Reviewer p9mz**
>
> Thanks for your comments.I will address your concerns point to point.
>
>
> 1. "Figures 3, 4 and 5 seem to be from other papers - please reference them properly." \
> Thanks for your suggestion.I've some of original images and cite the images from other papers in the newest version.
>
>
> 2. "Figure 6 is unclear in what it is trying to convey."\
> Thanks for your suggestion.I've deleted Figure 6 since it could not represents what I meant.
>
>
> 3. "(Very minor) Proper LaTeX for GIoU equation (use \cup for union, for example)"\
> Thanks for your suggestion.I've fixed some latex problems in the formula.
>
>
> 4. (Very minor) Please write out what NMS and WBF stands for before using the acronym, e.g. Weighted Box Function (WBF). This makes the paper easier to read for others.\
> Thanks for your suggestion.I've replaced NMS  with non-maximum impression and WBF with Weighted Box Function.
>
>
>
> 5. "Section 3 on Experiments and Section 4 on Results are very lacking in detail (especially Results)."\
> Thanks for your suggestion. I've added an efficiency comparison in section 4 and some result comparisons in section 2 as well.
>
>
> 6. "Overall, it is hard to verify whether the proposed model improves on existing models and the paper does not do a sufficient job in demonstrating the capabilities with respect to existing methods."\
> Thanks for your suggestion.  I've added a comparison between an existing method, Cellpose, and YUSEG in section 4.3.
>
>
>
> 7. "There is a clear limitation of the method when cells are crowded - a discussion of this in Limitations and proposal on how to overcome such problems would strengthen the paper."\
> Thanks for your suggestion. I agree with the limitation you stated. I've added discussion and potential solution to that limitation in section 4.5 "Limitation and future work".
>
>
> [1]. Stringer, Carsen, et al. "Cellpose: a generalist algorithm for cellular segmentation." Nature methods 18.1 (2021): 100-106.

---

### Official Review · Reviewer_uzoV · 2022-12-31
**Incomplete paper; therefore, it is hard to follow.**

**Rating:** 5
**Confidence:** 4

**Review:**

Summary and Contributions
The author proposed a simple yet efficient processing time by combining YOLO and U-net.

Strengths
- Providing sufficient details of the YoLo methods.
- Interesting empirical ideas to reduce the inference time.

Weaknesses:
- Lack of detail on the semantic segmentation task. In section 2.3: "the input images are the area that inside the bounding boxes," is that means each box obtained in the detection model is cropped to feed to the semantic model?
- The writing need to be checked and improved in the final version.  For example, the redundant words "Your idea Therefore" in the introduction part.
- Punctuation is misplaced in some places or missing.
- The quality of the figures needs to be improved as well, both in terms of the resolution and the content.

---

> ### Author Response · Authors · 2023-02-21
> **Response to Reviewer  Reviewer uzoV**
>
> Thanks for your advice.I will address your concerns point to point.
>
> 1."Lack of detail on the semantic segmentation task. In section 2.3: "the input images are the area that inside the bounding boxes,"
> Thanks for your advice. The detail of the semantic segmentation task is discussed in the section "Semantic segmentation of cell".The discussion included a diagram of the Unet structure, visualization of the training process , and a table that compare our model and others.
>
>
> 2."Is that mean each box obtained in the detection model is cropped to feed to the semantic model?"
> Yes. Your description is what happened in the inference stage . However, in the training process, the boxes are pre-calculated with the location information of the corresponding mask, which is the outer rectangle of each cell . Then each box is cropped to feed the semantic model.
>
>
> 3."The writing needs to be checked and improved in the final version. For example, the redundant words "Your idea Therefore" in the introduction part."
> Thanks for your advice. We updated our writing in the final version.
>
>
> 4."Punctuation is misplaced in some places or missing."
> Thanks for your advice. We found some punctuation is misplaced or missing. We fix them in the final version.
>
>
> 5."The quality of the figures needs to be improved as well, both in terms of the resolution and the content."
> Thanks for your advice. Yes, we agree the first version of the figures is poor. We updated some figures and add more high-quality figures.

---

### Decision · Program_Chairs · 2023-01-19

**Decision:**

Accept

**Comment:**

Although we accept the manuscript, please change fig4 to cell images.